# Rewriting History with Inverse RL: Hindsight Inference for Policy Improvement

**Benjamin Eysenbach**$^{*\phi\theta}$   **Xinyang Geng**$^{*\psi}$   **Sergey Levine**$^{\psi\theta}$   **Ruslan Salakhutdinov**$^{\phi}$

$^{\phi}$ Carnegie Mellon University   $^{\psi}$ UC Berkeley   $^{\theta}$ Google Brain

## Abstract

Multi-task reinforcement learning (RL) aims to simultaneously learn policies for solving many tasks. Several prior works have found that relabeling past experience with different reward functions can improve sample efficiency. Relabeling methods typically pose the question: if, in hindsight, we assume that our experience was optimal for some task, for what task was it optimal? Inverse RL answers this question. In this paper we show that inverse RL is a principled mechanism for reusing experience across tasks. We use this idea to generalize goal-relabeling techniques from prior work to arbitrary types of reward functions. Our experiments confirm that relabeling data using inverse RL outperforms prior relabeling methods on goal-reaching tasks, and accelerates learning on more general multi-task settings where prior methods are not applicable, such as domains with discrete sets of rewards and those with linear reward functions.

## 1   Introduction

Reinforcement learning (RL) aims to acquire control policies that take actions to maximize their reward, though existing RL algorithms remain data inefficient [11, 26]. Multi-task RL, where many RL problems are solved in parallel, has the potential to be more sample efficient than single-task RL, as data can be shared across tasks. Nonetheless, the problem of effectively sharing data across tasks remains largely unsolved.

The idea of sharing data across tasks has been studied at least since the 1990s [5]. More recently, a number of works have observed that retroactive relabeling of experience with different tasks can improve data efficiency [3, 24]. A common theme in prior relabeling methods is to relabel past trials with whatever goal or task was performed successfully in that trial. For example, in a goal-reaching task, we might use the state actually reached at the end of the trajectory as the relabeled goal, since the trajectory corresponds to a successful trial for the goal that was actually reached [3, 38]. However, prior work on goal relabeling is inapplicable to more general reward functions, such as discrete sets of reward functions or tasks defined by varying linear combinations of reward terms.

In this paper, we formalize prior relabeling techniques under the umbrella of *inverse* RL: by inferring the most likely task for a given trial via inverse RL, we provide a principled formula for relabeling in arbitrary multi-task problems. Inverse RL is *not* the same as evaluating a trajectory under all tasks and choosing whichever task yielded the highest reward. In fact, this strategy would often result in assigning most trajectories to the easiest task. Rather, inverse RL automatically takes into account the difficulty of each task by normalizing each reward function by the partition function. RL and inverse RL can be seen as complementary tools for maximizing reward: RL takes tasks and produces high-reward trajectories, and inverse RL takes trajectories and produces task labels such that the trajectories receive high reward. Formally, we prove that maximum entropy (MaxEnt) RL and MaxEnt inverse RL optimize the same multi-task objective: MaxEnt RL optimizes with respect

---

$^{*}$Equal contribution. Correspondence to beysenba@cs.cmu.edu

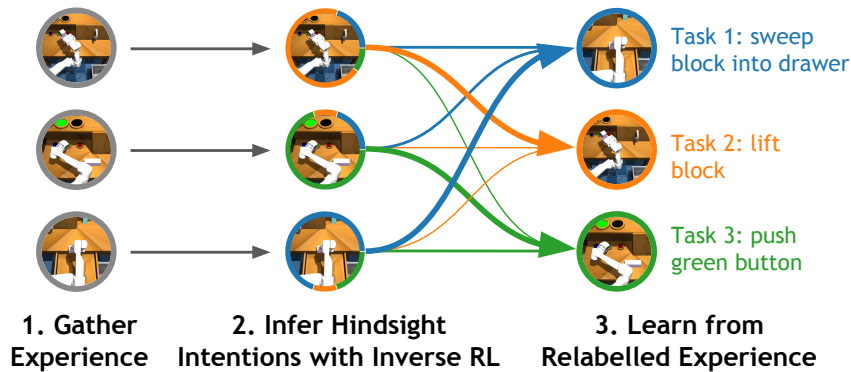

**1. Gather Experience**   **2. Infer Hindsight Intentions with Inverse RL**   **3. Learn from Relabelled Experience**

Figure 1: **Hindsight Inference for Policy Improvement (HIPI)**: Given a dataset of prior experience, we use inverse RL to infer the intentions of the agent's own past experience. We then use the relabeled experience with any policy learning algorithm, such as off-policy RL or supervised learning.

to trajectories, while MaxEnt inverse RL optimizes with respect to tasks. Unlike prior goal-relabeling techniques, we can use inverse RL to relabel experience for arbitrary task distributions, including linear or discrete reward sets. This observation suggests that RL and inverse RL might be combined to efficiently solve many tasks simultaneously. The combination we develop, Hindsight Inference for Policy Improvement (HIPI), first relabels experience with inverse RL and then uses the relabeled experience to learn a *task-conditioned* policy (see Fig. 1). One variant of this framework follows the same design as prior value-based goal-relabeling methods [3, 24, 38] but uses inverse RL to relabel experience, a difference that allows our method to handle arbitrary task families. The second variant has a similar design to self-imitation behavior cloning methods [15, 33, 45]: we relabel past experience using inverse RL and then learn a policy via task-conditioned behavioral cloning. Both algorithms are probabilistic reinterpretations and generalizations of prior work.

The main contribution of our paper is the observation that inverse RL is a principled mechanism for reusing experience across tasks. This observation not only provides insight into success of prior relabeling methods, but it also provides guidance on applying relabeling to arbitrary multi-task RL problems. Our second contribution is two simple algorithms that use inverse RL-based relabeling to accelerate multi-task RL. These algorithms *do not* require expert demonstrations, but rather perform inverse RL on the agent's own (possibly-random) past experience. Our experiments on complex simulated locomotion and manipulation tasks demonstrate that our approach outperforms state-of-the-art multi-task RL methods.

## 2   Related Work

We focus on multi-task RL problems, for which a number of algorithms have been proposed over the past decades [12, 22, 43, 51, 53]. Existing approaches still struggle to reuse data across multiple tasks, with researchers often finding that training separate models is a very strong baseline [58]. Similarly, prior work [16, 34, 44, 51] has effectively used independently-trained models to initialize multi-task models. When applying off-policy RL in the multi-task setting, a common technique is to take data collected when performing task A and pretend that it was collected for task B by recomputing the rewards at each step, effectively inflating the amount of data available for learning [3, 24, 38, 47]. In this paper we will show that this relabeling can be understood as inverse RL.

While RL asks how to go from a reward function to a policy, inverse RL asks the opposite question: after observing an agent acting in an environment, can we infer which reward function the agent was trying to optimize? Prior work has proposed a number of inverse RL algorithms [1, 41], with MaxEnt inverse RL being one of the most commonly used frameworks [14, 23, 62]. MaxEnt inverse RL can be viewed as the problem of inferring the posterior distribution over reward functions. [62] While most prior work uses maximum a-posteriori estimates, we follow prior work [7, 21, 40] in using the full posterior. Section 3 discusses how MaxEnt RL and MaxEnt inverse RL are related, with one problem being the dual of the other. It is therefore not a coincidence that many MaxEnt inverse RL algorithms involve solving a MaxEnt RL problem in the inner loop. Our method will do the opposite, using MaxEnt *inverse* RL in the inner loop of MaxEnt RL.

Our work builds on the idea that MaxEnt (forward) RL can be viewed as probabilistic inference. This idea has been proposed in a number of prior works [25, 28, 42, 52, 54–56] and used to construct practical RL algorithms [2, 18, 19]. While prior work [37, 60] has combined hindsight relabeling with MaxEnt RL, we will show that MaxEnt RL itself suggests a natural relabeling mechanism. Concurrent with our work, Li et al. [30] also propose to use inverse RL to relabel experience for off-policy RL. In contrast with this concurrent work, our analysis provides a principled unification of (forward) MaxEnt RL and MaxEnt inverse RL. We go further and observe that prior relabeling methods are *already* doing inverse RL, albeit as a special case, and present value-based and behavioral cloning-based variants of our method.

## 3 Preliminaries

This section reviews MaxEnt RL and MaxEnt inverse RL. We start by introducing notation.

**Notation.** We will analyze an MDP with states $s_t$, actions $a_t$, and a reward function $r(s_t, a_t)$. We sample actions from a policy $q(a_t \mid s_t)$. The initial state is sampled $s_1 \sim p_1(s_1)$ and subsequent transitions are governed by a dynamics distribution $s_{t+1} \sim p(s_{t+1} \mid s_t, a_t)$. We define a trajectory as a sequence of states and actions, $\tau = (s_1, a_1, \cdots)$, and write the likelihood of $\tau$ under policy $q$ as

$$q(\tau) = p_1(s_1) \prod_t p(s_{t+1} \mid s_t, a_t) q(a_t \mid s_t).$$

In the multi-task setting, we use $\psi \in \Psi$ to identify each task and assume that the prior $p(\psi)$ over tasks is known. The set of tasks $\Psi$ can be continuous or discrete, finite or infinite; each particular task $\psi \in \Psi$ can be continuous or discrete valued. We define $r_\psi(s_t, a_t)$ as the reward function for task $\psi$. Our experiments use both goal-reaching tasks, where $\psi$ is a goal state and $r_\psi$ is a distance metric, as well as more general task distributions, where $\psi$ specifies the hyperparameters of the reward function.

**MaxEnt RL.** MaxEnt RL maximizes the entropy-regularized sum of rewards [61]. MaxEnt RL can be equivalently expressed as a distribution matching problem. First, construct a reward-based *target distribution* $p(\tau)$ over trajectories,

$$p(\tau) \triangleq \frac{1}{Z} p_1(s_1) \prod_t p(s_{t+1} \mid s_t, a_t) e^{r(s_t, a_t)}. \tag{1}$$

Then, minimizing the reverse KL between the policy's trajectory distribution $q(\tau)$ and the target trajectory distribution $p(\tau)$ is equivalent to maximizing the entropy-regularized sum of rewards:

$$\max_q -D_{\mathrm{KL}}(q(\tau) \parallel p(\tau)) = \mathbb{E}_q\left[\left(\sum_t r(s_t, a_t) - \log q(a_t \mid s_t)\right) - \log Z\right], \tag{2}$$

The partition function $Z$ is introduced to make $p(\tau)$ integrate to one. Although the partition function is independent of the policy, and prior RL algorithms have therefore ignored it, it will play a crucial role in relabeling experience. Section 4 will examine the multi-task version of this objective, where rewards will depend on the task $\psi$.

**MaxEnt inverse RL.** Inverse RL observes previously-collected data and attempts to infer which reward function $r_\psi$ the actor was trying to maximize. While many inverse RL algorithms also yield a policy which is optimal for the inferred reward function, we will use "inverse RL" to refer to just the problem of inferring the reward. MaxEnt inverse RL [62] is a variant of inverse RL that defines the probability of trajectory $\tau$ being produced for task $\psi$ as

$$p(\tau \mid \psi) = \frac{p_1(s_1) \prod_t p(s_{t+1} \mid s_t, a_t) e^{r_\psi(s_t, a_t)}}{Z(\psi)}, \quad \text{where } Z(\psi) \triangleq \int p_1(s_1) \prod_t p(s_{t+1} \mid s_t, a_t) e^{r_\psi(s_t, a_t)} d\tau. \tag{3}$$

Applying Bayes' rule, the posterior distribution over tasks $\psi$ is given as follows:

$$p(\psi \mid \tau) = \frac{p(\tau \mid \psi) p(\psi)}{p(\tau)} \propto p(\psi) e^{\sum_t r_\psi(s_t, a_t) - \log Z(\psi)}. \tag{4}$$

In the next section, we will show that both MaxEnt RL and MaxEnt inverse RL minimize the same reverse KL divergence on the joint distribution of tasks and trajectories.

# 4 Hindsight Relabeling is Inverse RL

Off-policy RL allows us to use experience collected from one task to train the policy to solve another task. As one might expect, some trajectories are more useful for some tasks than others. Can we automatically determine for which tasks a given trajectory will be useful? Our analysis in the section suggests that one appealing answer is to apply MaxEnt inverse RL.

We start by defining a multi-task version of the MaxEnt RL objective (Eq. 2). In the multi-task setting, the target distribution $p(\tau, \psi)$ is a joint distribution over trajectories $\tau$ and tasks $\psi$. We can express the joint distribution as the product of a prior over tasks, $p(\psi)$, and the target trajectory distribution, $p(\tau \mid \psi)$ (Eq. 3):

$$p(\tau, \psi) = p(\psi) \frac{1}{Z(\psi)} p_1(s_1) \prod_t p(s_{t+1} \mid s_t, a_t) e^{r_\psi(s_t, a_t)}. \tag{5}$$

We can express the multi-task (MaxEnt) RL objective as the reverse KL divergence between the joint trajectory-task distributions:

$$\max_{q(\tau, \psi)} -D_{\mathrm{KL}}(q(\tau, \psi) \parallel p(\tau, \psi)). \tag{6}$$

If we factor the joint distribution as $q(\tau, \psi) = q(\tau \mid \psi) p(\psi)$, this objective is equivalent to maximizing the expected (entropy-regularized) reward of a task-conditioned policy $q(\tau \mid \psi)$:

$$\mathbb{E}_{\substack{\psi \sim p(\psi) \\ \tau \sim q(\tau \mid \psi)}} \left[ \left( \sum_t r_\psi(s_t, a_t) - \log q(a_t \mid s_t, \psi) \right) - \cancel{\log Z(\psi)} \right].$$

To share experience across tasks, we will want to figure out for which tasks a given trajectory will be useful. Thur, we instead choose to factor $q(\tau, \psi) = q(\psi \mid \tau) q(\tau)$, where $q(\tau)$ is represented non-parametrically as a distribution over previously-observed trajectories. The distribution $q(\psi \mid \tau)$ tells us the tasks for which trajectory $\tau$ is similar to the optimal policy for task $\psi$. We therefore expect that trajectory $\tau$ will be most useful for learning tasks $\psi$ where $q(\psi \mid \tau)$ is large. To compute the relabeling distribution, we expand our objective (Eq. 6) using this factorization:

$$\mathbb{E}_{\substack{\tau \sim q(\tau) \\ \psi \sim q(\psi \mid \tau)}} \left[ \log p_1(s_1) + \sum_t r_\psi(s_t, a_t) + \log p(s_{t+1} \mid s_t, a_t) + \log p(\psi) - \log q(\psi \mid \tau) - \log q(\tau) - \log Z(\psi) \right].$$

The relabeling distribution $q(\psi \mid \tau)$ that optimizes this objective is

$$q(\psi \mid \tau) \propto p(\psi) \exp\left( \sum_t r_\psi(s_t, a_t) - \log Z(\psi) \right). \tag{7}$$

Intuitively, we should use trajectory $\tau$ for solving tasks where the trajectory receives the highest reward, *as compared with the partition function*. The key observation here is that *the optimal relabeling distribution corresponds exactly to the MaxEnt inverse RL posterior over tasks* (Eq. 4). While the optimal relabeling distribution derived here depends on the entire trajectory, Appendix B shows how to perform relabeling when given a transition rather than an entire trajectory:

$$q(\psi \mid s_t, a_t) \propto p(\psi) \exp\left( \widetilde{Q}^q(s_t, a_t) - \log Z(\psi) \right) \tag{8}$$

In the next section we show that prior goal-relabeling methods are a special case of inverse RL.

## 4.1 Special Case: Goal Relabeling

A number of prior works have explicitly [3, 24, 38] and implicitly [15, 32, 45] found that hindsight relabeling can accelerate learning for *goal-reaching* tasks, where tasks $\psi$ correspond to goal states. We now show that these prior methods are a special case of inverse RL. Define a goal-conditioned reward function that penalizes the agent for failing to reaching the goal at the last step:

$$r_\psi(s_t, a_t) = \begin{cases} -\infty & \text{if } t = T \text{ and } s_t \neq \psi \\ 0 & \text{otherwise} \end{cases}. \tag{9}$$

We assume that the time step $t$ is included in $s_t$ to ensure that this reward function is Markovian. With this reward function, the optimal relabeling distribution $q(\psi \mid \tau)$ from Eq. 7 is $q(\psi \mid \tau) = \mathbb{1}(\psi = s_T)$,

where $s_T$ is the final state in trajectory $\tau$. Thus, *relabeling with the state actually reached is equivalent to inverse RL* when using the reward function in Eq. 9. While inverse RL is convenient when using this reward function, we rarely care about a policy's expected reward under this particular reward function. Viewing goal relabeling as a special case of inverse RL lets us extend goal relabeling to arbitrary task distributions. We will show that inverse RL handles task distributions including goal-reaching, discrete sets of tasks, and linear reward functions.

## 4.2 The Importance of the Partition Function

The partition function used by inverse RL is important for hindsight relabeling as it normalizes the rewards from tasks with varying difficulty and reward scale. Fig. 2 shows a didactic example with two tasks, where task $\psi_1$ is easier than task $\psi_2$, providing larger rewards to all trajectories. Relabeling with the task under which the agent received the largest reward (akin to Andrychowicz et al. [3]) fails because all experience will be relabeled with the first (easier) task. Subtracting the partition function from the rewards (as in Eq. 7) results in the desired behavior: trajectory $\tau_1$ is assigned task $\psi_1$ and $\tau_2$ is assigned to $\psi_2$.

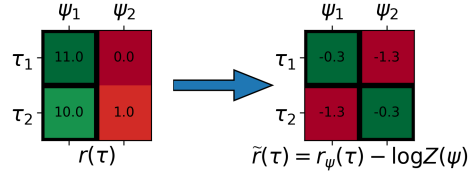

Figure 2: **The partition function normalizes rewards of different scales**: Two trajectories are evaluated on tasks with different reward scales. Black borders indicate the task to which we assign each trajectory. *(Left)* Without normalization, both trajectories are assigned to task $\psi_1$. *(Right)* After normalizing with the partition function, as is done by inverse RL (our method), trajectory $\tau_1$ is assigned task $\psi_1$ and $\tau_2$ is assigned to $\psi_2$.

## 4.3 How Much Does Relabeling Help?

Up to now, we have shown that the optimal way to relabel data is via inverse RL. We now obtain a lower bound on the improvement from relabeling. Both lemmas in this section will assume that a joint distribution $q(\tau, \psi)$ over tasks and trajectories be given (e.g., specified by a policy $q(a_t \mid s_t, \psi)$). We will define $q_\tau(\tau) = \int q(\tau, \psi)$ as the marginal distribution over trajectories and $q_\tau(\psi \mid \tau)$ as the corresponding optimal relabeling distribution (Eq. 7). Our aim is to show that the joint distribution after relabeling, $q_\tau(\tau, \psi) = q_\tau(\psi \mid \tau)q_\tau(\tau)$, is better than the joint distribution before relabeling, $q(\tau, \psi)$. We first show that relabeling data using inverse RL improves the MaxEnt RL objective:

**Lemma 1.** *The relabeled distribution $q_\tau(\tau, \psi)$ is closer to the target distribution than the original distribution, as measured by the KL divergence:*

$$D_{\mathrm{KL}}(q_\tau(\tau, \psi) \parallel p(\tau, \psi)) \leq D_{\mathrm{KL}}(q(\tau, \psi) \parallel p(\tau, \psi)).$$

*Proof.* Of the many possible relabeling distributions, one choice is to do no relabeling, assigning to each trajectory $\tau$ the task $\psi$ that was commanded when the trajectory was collected. Denote this relabeling distribution $q_0(\psi \mid \tau)$, so $q_0(\psi \mid \tau)q_\tau(\tau) = q(\tau, \psi)$. Because $q_\tau(\psi \mid \tau)$ minimizes the KL among all relabeling distributions (including $q_0(\psi \mid \tau)$), the desired inequality holds:

$$D_{\mathrm{KL}}(q_\tau(\psi \mid \tau)q_\tau(\tau) \parallel p(\tau, \psi)) \leq D_{\mathrm{KL}}(q_0(\psi \mid \tau)q_\tau(\tau) \parallel p(\tau, \psi)). \qquad \square$$

Thus, the relabeled data is an improvement over the original data, achieving a larger entropy-regularized reward (Eq. 6). As our experiments will confirm, relabeling data will accelerate learning. Our next result will give us a lower bound on the size of this improvement:

**Lemma 2.** *The improvement in the MaxEnt RL objective (Eq. 6) gained by relabeling is lower bounded as follows:*

$$D_{\mathrm{KL}}(q(\tau, \psi) \parallel p(\tau, \psi)) - D_{\mathrm{KL}}(q_\tau(\tau, \psi) \parallel p(\tau, \psi)) \geq \mathbb{E}_{q_\tau}\left[D_{\mathrm{KL}}(q(\psi \mid \tau) \parallel q_\tau(\psi \mid \tau))\right].$$

The proof, a straightforward application of information geometry, is in Appendix A. This result says that the amount that relabeling helps is at least as large as the difference between the original task labels $q(\psi \mid \tau)$ and the task labels inferred by inverse RL, $q_\tau(\psi \mid \tau)$. When experience is collected from a random policy, there is little correlation between the originally-commanded tasks and the tasks inferred by inverse RL, so we expect a large gain from relabeling. Experience from the optimal policy is already optimally labeled, so the improvement from relabeling drops to zero once we have acquired the optimal policy.

**Algorithm 1** Approximate Inverse RL.
When used in HIPI-RL (Alg. 2) we only have transitions, so we compute $R_{\psi^{(j)}}^{(i)}$ using Eq. 8 (blue line). When used in HIPI-BC (Alg. 3) we have full trajectories, so we compute $R_{\psi^{(j)}}^{(i)}$ using Eq. 7 (red line).

---

**function** INVERSERL($\{(s_t^{(i)}, a_t^{(i)}, s_{t+1}^{(i)}, \psi^{(i)})\}$)
  **for** $j = 1, \cdots, B$ **do**      ▷ task index
    **for** $i = 1, \cdots, B$ **do**    ▷ state-action index
      $R_{\psi^{(j)}}^{(i)} \leftarrow \widetilde{Q}(s^{(i)}, a^{(i)}, \psi^{(j)})$     ▷ Eq. 8
      $R_{\psi^{(j)}}^{(i)} \leftarrow \sum_t r_{\psi^{(j)}}(s_t^{(i)}, a_t^{(i)})$     ▷ Eq. 7
    $\log Z(\psi^{(j)}) \leftarrow \frac{1}{B}\sum_{i=1}^{B} e^{R_{\psi^{(j)}}^{(i)}}$
  **for** $i = 1, \cdots, B$ **do**
    $\widetilde{\psi}^{(i)} \sim$ SOFTMAX($R_{\psi^{(1)}}^{(i)} - \log Z(\psi^{(1)}), \cdots$)
  **return** $\{\widetilde{\psi}^{(i)}\}$

---

**Algorithm 2** HIPI-RL:
Inverse RL for Off-Policy RL

---

**while** not converged **do**
  $\{(s_t^{(i)}, a_t^{(i)}, s_{t+1}^{(i)}, \psi^{(i)})\} \sim$ REPLAYBUFFER
  $\{\widetilde{\psi}^{(i)}\} \leftarrow$ INVERSERL($\{(s_t^{(i)}, a_t^{(i)}, s_{t+1}^{(i)}, \psi^{(i)})\}$)
  $\widetilde{Q} \leftarrow$ MAXENT RL($\{(s_t^{(i)}, a_t^{(i)}, s_{t+1}^{(i)}, \widetilde{\psi}^{(i)})\}$)

---

**Algorithm 3** HIPI-BC:
Inverse RL for Behavior Cloning

---

**while** not converged **do**
  $\{(s_t^{(i)}, a_t^{(i)}, s_{t+1}^{(i)}, \psi^{(i)})\} \sim$ REPLAYBUFFER
  $\{\widetilde{\psi}^{(i)}\} \leftarrow$ INVERSERL($\{(s_t^{(i)}, a_t^{(i)}, s_{t+1}^{(i)}, \psi^{(i)})\}$)
  $\theta \leftarrow \theta + \eta \nabla_\theta \sum_i \log \pi_\theta\left(a_t^{(i)} \mid s_t^{(i)}, \widetilde{\psi}^{(i)}\right)$
  **return** $\pi_\theta$

## 5 Using Inverse RL to Accelerate RL

In this section we propose two multi-task policy search algorithms that use inverse RL to relabel experience. We will call both methods *Hindsight Inference for Policy Improvement (HIPI)*. Both methods will follow the same general recipe. Inverse RL (Alg. 1) takes as input transitions or trajectories and returns a distribution over tasks. Our policy search algorithms (Algorithms 2 and 3) take past experience, use inverse RL to sample a corresponding task, and then update the policy using experience together with the task label. The two algorithms differ in how they update the policy: HIPI-RL (Sec. 5.1) will use off-policy MaxEnt RL, whereas HIPI-BC (Sec. 5.2) will use behavior cloning. Full experimental details are included in Appendix E and code has been released.[2]

### 5.1 Using Relabeling Data for Off-Policy RL (HIPI-RL)

HIPI-RL is a multi-task RL algorithm that uses experience relabeled with inverse RL for off-policy RL. We describe the algorithm in Alg. 2. At each iteration, we sample a batch of $B$ transitions from the replay buffer. For each transition in the batch, we run inverse RL, sample a new task $\widetilde{\psi}$ from the inverse RL posterior, and *relabel* that transition to be used for learning task $\widetilde{\psi}$. We then perform MaxEnt RL on the batch of relabeled transitions. We use SAC [19] as the underlying MaxEnt RL algorithm. The only difference between HIPI-RL and SAC is that HIPI-RL relabels experience with inverse RL before applying the standard SAC updates. To collect new experience, we sample a task from the prior, $\psi \sim p(\psi)$, and then take actions using corresponding (stochastic) policy, $\pi(a \mid s, \psi)$. HIPI-RL can be viewed as a generalization of prior relabeling techniques [3, 24], allowing them to be applied to task distributions beyond goal-reaching.

The key design decision is the choice of inverse RL algorithm. Alg. 1 outlines one approximate inverse RL algorithm that can be efficiently integrated into off-policy RL. The algorithm takes as input $B$ transitions $\{(s_t^{(i)}, a_t^{(i)}, s_{t+1}^{(i)})\}$ along with the tasks that were commanded when these transitions were collected, $\{\psi^{(i)}\}$. Rather than consider all (possibly-infinite) tasks, we make a non-parametric approximation and only consider the likelihood of these $B$ originally-commanded tasks. Then, for all pairs $1 \leq i, j \leq B$, we compute the likelihood that transition $(s_t^{(i)}, a_t^{(i)}, s_{t+1}^{(i)})$ is optimal for task $\psi^{(j)}$ following Eq. 8. Finally, we normalize these likelihoods by taking a softmax, which is equivalent to using the following (biased) approximation of the partition function $\log Z(\psi)$:

$$\log Z(\psi) = \log \int e^{R_\psi(s,a)} ds\, da \approx \log \frac{1}{B} \sum_{i=1}^{B} e^{R_\psi(s^{(i)}, a^{(i)})} + \log B.$$

Alg. 1 is one of many possible methods for inverse RL. Alternative methods include doing gradient descent on the tasks $\psi$ or learning a parametric task-sampler to approximate the optimal relabeling distribution (Eq. 7). We leave this as future work.

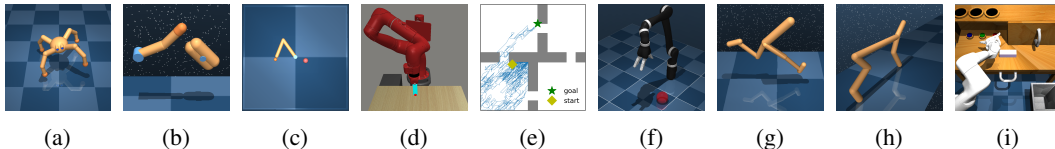

| (a) | (b) | (c) | (d) | (e) | (f) | (g) | (h) | (i) |

Figure 4: **Environments for experiments**: *(a)* quadruped, *(b)* finger, *(c)* 2D reacher, *(d)* sawyer reach, *(e)* 2D navigation *(f)* jaco reach, *(g)* walker, *(h)* cheetah, and *(i)* desk manipulation.

## 5.2 Using Relabeled Data for Behavior Cloning (HIPI-BC)

Our second multi-task policy search algorithm, HIPI-BC, uses behavior cloning to learn from relabeled past data. We describe the algorithm Alg. 3. Similar to HIPI-RL, each iteration samples a batch of experience and relabels that experience with inverse RL. Rather than using this experience for RL, HIPI-BC directly performs behavior cloning, maximizing the following objective:

$$\max_\theta \sum_{i=1}^{B} \log \pi_\theta(a_t^{(i)} \mid s_t^{(i)}, \widetilde{\psi}^{(i)})$$

In Appendix C, we show that HIPI-BC generalizes a number of previous methods, extending variational policy search [10, 29, 35, 36] to the multi-task setting and extending goal-conditioned imitation learning [15, 32, 45] to arbitrary task distributions. Our implementation uses Alg. 1 for inverse RL. We will use a trajectory-level replay buffer, so we can directly compute the reward of each trajectory under each task, rather than approximating the future rewards with the Q function. See Appendix E.2 for hyperparameters.

## 6 Experiments: Relabeling with Inverse RL Accelerates Learning

Our experiments focus on two methods for using relabeled data: off-policy RL (Alg. 2) and behavior cloning (Alg. 3). We evaluate our method on both goal-reaching tasks as well as more general task distributions, including linear combinations of a reward basis and discrete sets of tasks. The aim of all experiments is to maximize the task reward, not to imitate an expert.

### 6.1 HIPI-RL: Inverse RL for Off-Policy RL

Our first set of experiments apply Alg. 2 to domains with varying reward structure, demonstrating how relabeling data with inverse RL can accelerate off-policy RL.

**Didactic example.** We start with a didactic example to motivate why relabeling experience with inverse RL should accelerate off-policy RL. In the gridworld shown in Fig. 3, we construct a dataset with two trajectories: $A \to B$ and $C \to D$. From state A, inverse RL identifies many possible intentions, including states $B$ and $D$, so both $A \to B$ and $A \to D$ get included in the relabeled data. In contrast, final state relabeling (HER) only uses trajectory $A \to B$. We then apply (soft) Q-learning to both datasets. Whereas Q-learning with final state relabeling only succeeds at reaching those goals in the top row (6/10 goals), our approach, which corresponds to Q-learning with inverse RL, relabeling succeeds at reaching all goals. Note that relabeling with future states would also fail to use states from trajectory $C \to D$ as goal state

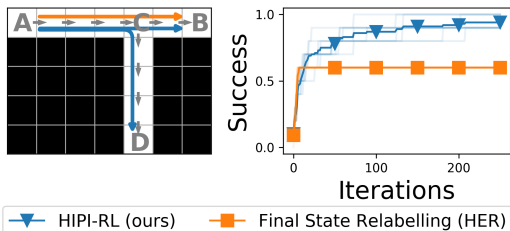

Figure 3: **Relabeling stitches crossing trajectories**: *(Left)* A gridworld with two observed trajectories $A \to B$ and $C \to D$. Inverse RL identifies both $B$ and $D$ as likely intentions from state $A$ and includes both $A \to B$ and $A \to D$ in the relabeled data. Final state relabeling (HER) only relabels with the goal actually achieved, corresponding to trajectory $A \to B$. *(Right)* HIPI-RL learns to reach all goals, whereas HER only learns to reach the 6/10 goals along the top row.

$A$. The remainder of this section will show the benefits of relabeling using inverse RL in domains of increasing complexity.

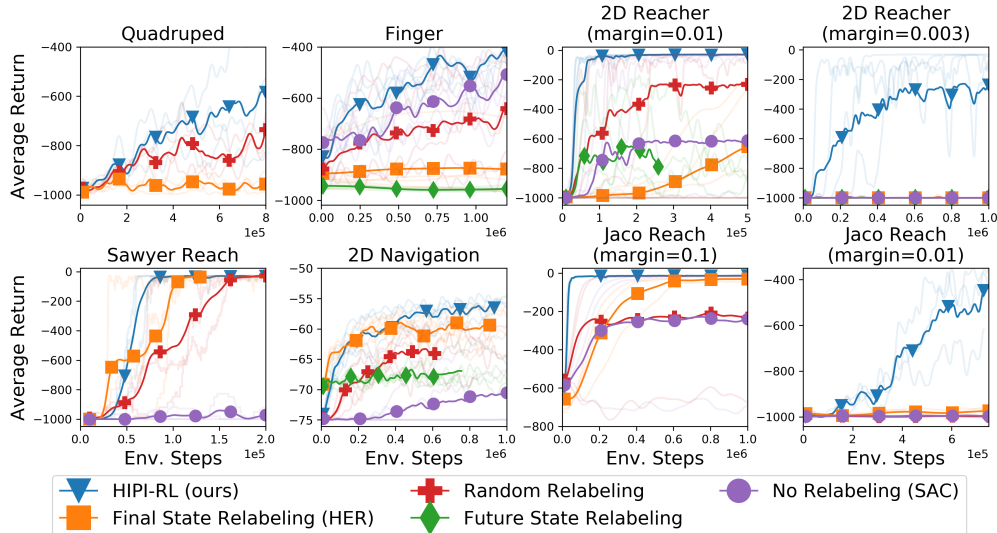

Figure 5: **Relabeling for goal-reaching tasks**: On six goal-reaching domains, relabeling with inverse RL (our method) learns faster than with previous relabeling strategies. On extremely sparse versions of two tasks, shown in the right column, only our method learns the tasks. See text for details.

**Goal-reaching task distributions.** We next apply our method to goal-reaching tasks, where each task $\psi$ corresponds to reaching a different goal state. We used six domains, shown in Fig. 4: a quadruped locomotion task, a robotic finger turning a knob, a 2D reacher, a reaching task on the Sawyer robot, a 2D navigation environment with obstacles, and a reaching task on the Jaco robot. Appendix E provides details of all tasks. We compared our method against four alternative relabeling strategies: relabeling with the final state reached (HER [3]), relabeling with a randomly-sampled task, relabeling with a future state in the same trajectory, and doing no relabeling (SAC [19]). For tasks where the goal state only specifies certain dimensions of the state, relabeling with the final state and future state requires privileged information indicating which state dimensions correspond to the goal. For example, in the quadruped domain, these methods need to be told that the agent's center of mass is stored in state dimensions 0 and 1. Our method does not require this additional information. We found that the most sensitive parameter was the number of gradient steps per environment step. We tuned this parameter independently for each method, and report the best results for each method.

As shown in Fig. 5, relabeling experience with inverse RL (our method) always learns at least as quickly as the other relabeling strategies, and often achieves larger asymptotic reward. While final state relabeling (HER) performs well on some tasks, it is worse than random relabeling on other tasks. We also observe that random relabeling is a competitive baseline, provided that the number of gradient steps is sufficiently tuned. We conjectured that soft relabeling would be most beneficial in settings with extremely sparse rewards. To test this hypothesis, we modified the reward functions in 2D reacher and Jaco reaching environments to be much sparser. As shown in the far right column on Fig. 5, only soft relabeling is able to make learning progress in this setting.

**More general task distributions.** Our next experiment demonstrates that, in addition to relabeling goals, inverse RL can also relabel experience for more general tasks distributions. Our first task distribution is a discrete set of goal states $\psi \in \{1, \cdots, 32\}$ for the 2D reacher environment. The second task distribution highlights the capability of inverse RL to relabel experience for classes of reward functions defined as linear combinations $r_\psi(s, a) = \sum_{i=1}^{d} \psi_i \phi_i(s, a)$ of features

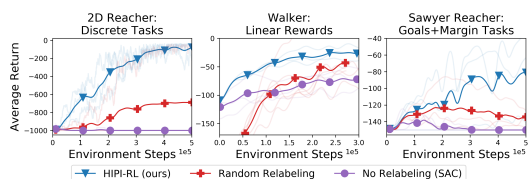

Figure 6: **Relabeling for general tasks distributions**: On all tasks, relabeling with inverse RL accelerates learning and leads to larger asymptotic reward. Existing relabeling strategies are not applicable in this setting.

$\phi(s, a) \in \mathbb{R}^d$. We use the walker environment, with features corresponding to torso height, velocity, relative position of the feet, and a control cost. The third task distribution is again a goal reaching task, but one where the task $\phi = (s_g, m)$ indicates both the goal state as well as the desired margin from that goal state. As prior relabeling approaches are not applicable to these general task distributions, we

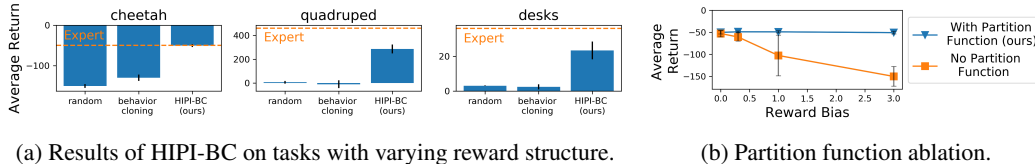

(a) Results of HIPI-BC on tasks with varying reward structure.  (b) Partition function ablation.

Figure 7: **HIPI-BC**: *(Left)* Behavior cloning on experience relabeled with inverse RL boosts reward on goal-reaching tasks, linear reward functions, and discrete tasks. *(Right)* Removing the partition function from inverse RL results in poor performance.

only compared our approach to random relabeling and no relabeling. As shown in Fig. 6, relabeling with inverse RL provides more sample efficient learning in all tasks, and the asymptotic reward is larger than the baselines by a non-trivial amount in two of the three tasks.

## 6.2 HIPI-BC: Behavioral Cloning on Experience Relabeled with Inverse RL

We now present experiments using HIPI-BC (Alg. 3), performing behavior cloning on the experience relabeled with inverse RL. We use three domains with varying types of rewards: (1) half-cheetah with continuous goal velocities; (2) quadruped with linear reward functions; and (3) manipulation with nine discrete tasks. For the half-cheetah and quadruped domains, we collected 1000 demonstrations from a policy trained with off-policy RL. For the manipulation environment, Lynch et al. [32] provided a dataset of 100 demonstrations for each of these tasks, which we aggregate into a dataset of 900 demonstrations. In all settings, we discarded the task labels, simulating the common real-world setting where experience does not come prepared with task labels. As shown in Fig. 7a, first inferring the tasks with inverse RL and then performing behavioral cloning results in significantly higher final rewards than task-agnostic behavior cloning on the entire dataset, which is no better than random.

Our final experiment demonstrates the importance of the partition function. On the cheetah domain, we synthetically corrupt the demonstrations by adding a constant bias to the reward for the first task. We then compare the performance of our approach against an ablation that does not normalize by the partition function when relabeling data. As shown in Fig. 7b, the performance of this ablation degrades as the reward bias increases, whereas our method, which normalizes the task rewards in the inverse RL step, is not affected.

## 7 Discussion

In this paper, we proved that inverse RL is as a principled mechanism for sharing experience across tasks. We showed that a number of prior works can be understood as special cases of this framework. The idea that inverse RL might be used to relabel data is powerful because it enables us to extend relabeling techniques to general classes of reward functions. We used this idea to propose two multi-task policy search algorithms, which relabel past experience with inverse RL and use this relabeled experience for off-policy RL and supervised learning.

We are only scratching the surface of the many ways relabeled experience might be used to accelerate learning. For example, the problem of task inference is ever-present in meta-learning, and it is intriguing to imagine explicitly incorporating inverse RL into meta RL. Broadly, we hope that the observation that inverse RL can be used to accelerate RL will spur research on better inverse RL algorithms, which in turn will provide better RL algorithms.

**Limitations** Our algorithms do require that the user manually specify the *family* of reward functions, over which we perform inverse RL. If this family of reward functions is overly narrow, our algorithm will fail to learn omitted tasks. This failure mode can easily be mitigated by using expressive task families. Indeed, because we take a nonparametric approach to inverse RL (Alg. 1), the time complexity of our approach does not increase with the size of the task family. We conjecture that our algorithm will work better in settings with highly expressive task families, as increasing the number of reward functions increases the likelihood that a given trajectory is optimal for *some* task.

Inverse RL, which our approach uses as a building block, remains a challenging problem. We found in our experiments that in cases where inverse RL failed, it returned a uniform relabeling distribution $q(\psi \mid \tau)$. Thus, when inverse RL fails, our method resorts to (uniform) random relabeling. As shown in Fig. 5 and Fig. 6, random task relabeling is surprisingly effective, suggesting that such a failure case is not overly problematic.

## Broader Impact

In this paper, we showed that hindsight relabeling is a form of inverse RL and used this insight to propose algorithms which can effectively share experience for solving multiple tasks. These algorithms may prove valuable in scenarios where data collection is costly or dangerous. Additionally, the use of inverse RL makes our algorithm robust to misspecification in the scale of the reward function (see Sec. 4.2 and Fig. 7b); an adversary *cannot* bias our algorithm to learn a certain task by scaling the reward for that task.

Today, sharing data across tasks remains challenging, so users are forced to collect data anew when they want to solve new tasks. The result is that users with access to robots have an upper-hand in teaching robots to perform new tasks. If we were able to effectively share data across tasks, then this balance of power would shift towards owners of data, rather than owners of robots (though, in many cases, these are one and the same). Indeed, this seems to have been the trend in supervised learning [48]. This risk might be mitigated by sharing data across institutions, as is starting to be done for robot manipulation [9] and autonomous vehicles [4, 6, 27, 46, 49, 57].

## Acknowledgments and Disclosure of Funding

We thank Yevgen Chebotar, Aviral Kumar, Vitchyr Pong, and Anirudh Vemula for formative discussions. We are grateful to Ofir Nachum for pointing out the duality between MaxEnt RL and the partition function, and to Karol Hausman for reviewing an early draft of this paper. We thank the anonymous NeurIPS reviewers for useful feedback. We thank Stephanie Chan, Corey Lynch, and Pierre Sermanet for providing the desk manipulation environment. This research was supported by the Fannie and John Hertz Foundation, NASA, DARPA, US Army, and the National Science Foundation (IIS-1700696, IIS-1700697, IIS1763562, and DGE 1745016). Any opinions, findings and conclusions or recommendations expressed in this material are those of the authors and do not necessarily reflect the views of the National Science Foundation.

## Footnotes

[2]`https://github.com/google-research/google-research/tree/master/hipi`

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
