[Supplementary Material]

# A  Proof of Lemma 2

This section provides a proof of Lemma 2.

*Proof.* The optimal relabeling distribution can be viewed as an information projection of the joint distribution $q_\tau(\psi \mid \tau)q_\tau(\tau)$ onto the target distribution $p(\tau, \psi)$ (Eq. 5):

$$q_\tau(\psi \mid \tau)q_\tau(\tau) = \min_{q_\tau \in \mathcal{Q}_\tau} D_{\mathrm{KL}}(q_\tau(\psi \mid \tau)q_\tau(\tau, \psi) \parallel p(\tau, \psi)),$$

where $\mathcal{Q} = \{q(\tau, \psi) \text{ s.t. } \int q(\tau, \psi)d\psi = q_\tau(\tau)\}$ is the set of all joint distributions with marginal $q_\tau(\tau)$. Note that this set $\mathcal{Q}$ is closed and convex. We then apply Theorem 11.6.1 from Cover and Thomas [8]:

$$D_{\mathrm{KL}}(q(\tau, \psi) \parallel p(\tau, \psi)) \geq D_{\mathrm{KL}}(q_\tau(\tau, \psi) \parallel p(\tau, \psi)) + D_{\mathrm{KL}}(q(\tau, \psi) \parallel q_\tau(\tau, \psi)). \quad (10)$$

The second KL divergence on the RHS can be simplified:

$$D_{\mathrm{KL}}(q(\tau, \psi) \parallel q_\tau(\tau, \psi)) = D_{\mathrm{KL}}(q(\psi \mid \tau)q_\tau(\tau) \parallel q_\tau(\psi \mid \tau)q_\tau(\tau))$$
$$= \underline{D_{\mathrm{KL}}(q_\tau(\tau) \parallel q_\tau(\tau))} + \mathbb{E}_{q_\tau}\left[D_{\mathrm{KL}}(q(\psi \mid \tau) \parallel q_\tau(\psi \mid \tau))\right]$$

Substituting this simplification into Eq. 10 and rearranging terms, we obtain the desired result. $\quad\square$

# B  Inverse RL on Transitions

For simplicity, our derivation of relabeling in Section 4 assumed that entire trajectories were provided. This section outlines how to do relabeling with inverse RL when we are only provided with $(s, a, s')$ transitions, rather than entire trajectories. This derivation will motivate the use of the *soft* Q-function in Eq. 8. In this case, policy distribution $q$ in the MaxEnt RL objective (Eq. 6) is conditioned on the current state and action $(s_t, a_t)$ in addition to the task $\psi$:

$$\max_{q(\tau, \psi \mid s_t, a_t)} -D_{\mathrm{KL}}(q(\tau, \psi \mid s_t, a_t) \parallel p(\tau, \psi)). \quad (11)$$

Following the derivation in Section 4, we expand this objective, using $q(\psi \mid s_t, a_t)$ as our relabeling distribution:

$$\mathbb{E}_{\substack{\psi \sim q(\psi \mid s_t, a_t) \\ \tau \sim q(\tau \mid \psi, s_t, a_t)}} \left[ \sum_{t'=t} r_\psi(s_{t'}, a_{t'}) + \underline{\log p(s_{t'+1} \mid s_{t'}, a_{t'})} - \log q(a_{t'} \mid s_{t'}, \psi) - \underline{\log p(s_{t'+1} \mid s_{t'}, a_{t'})} \right.$$
$$\left. + p(\psi) - \log q(\psi \mid s_t, a_t) - \log Z(\psi) \right]. \quad (12)$$

The expected value of the two summations is the *soft* Q-function for the policy $q(a \mid s, \psi)$:

$$\widetilde{Q}^q(s_t, a_t, \psi) = \mathbb{E}_{\substack{\psi \sim q(\psi \mid s_t, a_t) \\ \tau \sim q(\tau \mid \psi, s_t, a_t)}} \left[ \sum_{t'=t} r_\psi(s_{t'}, a_{t'}) - \log q(a_{t'} \mid s_{t'}, \psi) \right]. \quad (13)$$

Substituting Eq. 13 into Eq. 12 and ignoring terms that do not depend on $\psi$, we can solve the optimal relabeling distribution:

$$q(\psi \mid s_t, a_t) \propto p(\psi)e^{\widetilde{Q}^q(s_t, a_t, \psi) - \log Z(\psi)}. \quad (14)$$

**Learning Dynamics**  In practice, we estimate the partition function $Z(\psi)$ from previously-observed experience. If we have not seen any high-reward trajectories for a particular task $\psi$, then our approximate inverse RL method will underestimate the partition function $Z(\psi)$, causing mediocre trajectories to appear near-optimal for task $\psi$. We conjecture that this makes learning easier, providing allowing the agent to effectively learn before having observed optimal trajectories. We leave further investigation of this effect as future work.

| (a) Original gridworld. | (b) Modified gridworld where agent cannot move left. |

Figure 8: **Visualizing inverse RL**: We visualize the goals inferred by inverse RL on two gridworld domains. Each subplot corresponds to a different transition, indicated by the orange arrow. More likely goals are colored dark blue, while unlikely goals are colored white.

## C    Prior Methods are Special Cases of HIPI

Prior work on both goal-conditioned supervised learning, self-imitation learning, and reward-weighted regression can all be understood as special cases of HIPI-BC. Goal-conditioned supervised learning [15, 32, 45] learns a goal-conditioned policy using a dataset of past experience. For a given state, the action that was actually taken is treated as the correct action (i.e., label) for states reached in the future, and a policy is learned via supervised learning. As discussed in Section 4.1, relabeling with the goal actually achieved is a special case of our framework. We refer the reader to those papers for additional evidence that combining inverse RL (albeit a trivial special case) with behavior cloning can effectively learn complex control policies. Self-imitation learning [33] and iterative maximum likelihood training [31] augment RL with supervised learning on a handful of the best previously-seen trajectories, an approach that can be viewed in the inverse RL followed by supervised learning framework. However, because the connection to inverse RL is not made precise, these methods omit the partition function, which may prove problematic when extending these methods to multi-task settings. Finally, single-task RL methods based on variational policy search [28] and reward-weighted regression [35, 36] can also be viewed in this framework. Noting that the optimal relabeling distribution is given as $q(\psi \mid \tau) \propto \exp(R_\psi(\tau) - \log Z(\psi))$, relabeling by sampling from the inverse RL posterior and then performing behavior cloning can be written concisely as the following objective:

$$\int e^{R_\psi(\tau) - \log Z(\psi)} \sum_t \log \pi(a_t \mid s_t, \psi) d\psi d\tau.$$

The key difference between this objective and prior work is the partition function. The observation that these prior methods are special cases of inverse RL allows us to apply similar ideas to arbitrary classes of reward functions, a capability we showcase in our experiments.

## D    Additional Experiments

In this section we describe two additional experiments.

**Visualizing the inferred goals.**    Fig. 8 visualize the inferred goals on the gridworld example (Sec. 5.1). Each subplot corresponds to a different transition, denoted by the orange arrow. Dark blue cells denote likely goals, while white cells denote unlikely goals. When the dynamics are modified so the agent cannot move left (Fig. 8b), states to the left of the agent are no longer inferred as likely goals.

**Effect of batch size.**    We ran an additional experiment varying the batch size used by HIPI-RL on the sparse 2D reacher. Fig. 9 (right) shows that increasing the batch size significantly improves performance, suggesting that better approximate inverse RL results in better performance. We used a batch size of 32 for the results in the paper, but this experiment suggests that we could have gotten stronger results by using a larger batch size.

Figure 9: Varying the batch size on sparse 2D reacher

# E    Experimental Details

## E.1    Hyperparameters for Off-Policy RL

Except for the didactic experiment, we used SAC [19] as our RL algorithm, taking the implementation from Guadarrama et al. [17]. This implementation scales the critic loss by a factor of 0.5. Following prior work [38], we only relabeled 50% of the samples drawn from the replay buffer, using the originally-commanded task the remaining 50%. The only hyperparameter that differed across relabeling strategies was the number of gradient updates per environment step. For each experiment, we evaluated each method with values in $\{1, 3, 10, 30\}$ and reported the results of the best hyperparameter in our plots. Perhaps surprisingly, doing random relabeling but simply increasing the number of gradient updates per environment step was a remarkably competitive baseline.

- Learning Rate: 3e-4 (same for actor, critic, and entropy dual parameter)
- Batch Size: 32
- Network architecture: The input was the concatenation of the state observation and the task $\psi$. Both the actor and critic networks were 2 hidden layer ReLu networks. The actor output was squashed by a tanh activation to lie within the actor space constraints. There was no activation at the final layer of the critic network, except in the desk environment (see comment below). The hidden layer dimensions were (32, 32) for the 2D navigation environments, (256, 256) for the quadruped and desk environments, and (64, 64) for all other environments.
- Discount $\gamma$: 0.99
- Initial data collection steps: 1e5
- Target network update period: 1
- Target network $\tau$: 0.005
- Entropy coefficient $\alpha$: We used the entropy-constrained version of SAC [20], using $-\dim(\mathcal{A})$ as the target value, where $\dim(\mathcal{A})$ is the action space dimension.
- Replay buffer capacity: 1e6
- Optimizer: Adam
- Gradient Clipping: We found that clipping the gradients to have unit norm was important to get RL working on the Sawyer and Jaco tasks.

To implement final state relabeling, we modified transitions as they were being added to the replay buffer, adding both the original transition and the transition augmented to use the final state as the goal. To implement future state relabeling, we modified transitions as they were being added to the replay buffer, adding both the original transition and a transition augmented to use one of the next 4 states in the same trajectory as the goal.

## E.2    Hyperparameters for Behavior Cloning Experiments

To account for randomness in the learning process, we collect at least 200 evaluation episodes per domain; we repeat this experiment for at least 5 random seeds on each domain, and plot the mean and standard deviation over the random seeds. We used a 2-layer neural network with ReLu activations for all experiments. The hidden layers had size (256, 256). We optimized the network to minimize MSE using the Adam optimizer with a learning rate of 3e-4. We used early stopping, halting training when the validation loss increased for 3 consecutive epochs. Typically training converged in 30 - 50 epochs. We normalized both the states and actions. For the task-conditioned experiments, we concatenated the task vectors to the state vectors.

### E.3 Quadruped Environment

The quadruped was a modified version of the environment from Abdolmaleki et al. [2]. We modified the initial state distribution so the agent always started upright, and modified the observation space to include the termination signal as part of the observation. Tasks $\psi \in \mathbb{R}^2$ were sampled uniformly from the unit circle. Let $s_{\text{XY vel}}$ and $s_{\text{XY pos}}$ indicate the XY velocity and position of the agent. For the HIPI-RL experiments, we used the following sparse reward function:

$$r_\psi(s, a) = \mathbb{1}(\|s_{\text{XY pos}} - \psi\|_2 \leq 0.3) - 1.0,$$

and the episode terminated when $\|s_{\text{XY pos}} - \psi\|_2 \leq 0.3$. We also reset the environment after 300 steps if the agent had failed to reach the goal. For the HIPI-BC experiments, we used the following dense reward function:

$$r_\psi(s, a) = s_{\text{XY vel}}^T \psi + 0.1\|a\|_2^2.$$

Episodes were 300 steps long.

Quadruped

### E.4 Finger Environment

The finger environment was taken from Tassa et al. [50]. Tasks $\psi$ were sampled using the environment's default goal sampling function. Let $s_{\text{XY}}$ denote the XY position of the knob that the agent can manipulate. The reward function was defined as

$$r_\psi(s, a) = \mathbb{1}(\|s_{\text{XY}} - \psi\|_2 \leq 0.01) - 1.0$$

and the episode terminated when $\|s_{\text{XY}} - \psi\|_2 \leq 0.01$. We also reset the environment after 1000 steps if the agent had failed to reach the goal.

Finger

### E.5 2D Reacher Environment

The 2D reacher environment was taken from Tassa et al. [50]. Let $s_{\text{XY}}$ denote the XY position of the robot end effector. The reward function was defined as

$$r_\psi(s, a) = \mathbb{1}(\|s_{\text{XY}} - \psi\|_2 \leq m) - 1.0$$

and the episode terminated when $\|s_{\text{XY}} - \psi\|_2 \leq m$, where $m > 0$ is a margin around the goal. We used $m = 0.01$ and $m = 0.003$ in our experiments. We also reset the environment after 1000 steps if the agent had failed to reach the goal. Tasks were sampled using the environment's default goal sampling function.

2D Reacher

### E.6 Sawyer Reach Environment

The sawyer reach environment was taken from Yu et al. [59]. Let $s_{\text{XYZ}}$ denote the XYZ position of the robot end effector. The reward function was defined as

$$r_\psi(s, a) = \mathbb{1}(\|s_{\text{XYZ}} - \psi\|_2 \leq m) - 1.0$$

and the episode terminated when $\|s_{\text{XY}} - \psi\|_2 \leq m$, where $m > 0$ is a margin around the goal. We used $m = 0.01$ and $m = 0.003$ in our experiments. We also reset the environment after 150 steps if the agent had failed to reach the goal. Tasks were sampled using the environment's default goal sampling function. For the experiment where the task indicator $\psi$ also specified the margin $m$, the margin was sampled uniformly from the interval $[0, 0.1]$.

Sawyer Reach

### E.7 2D Navigation Environment

We used the 2D navigation environment from Eysenbach et al. [13]. The action space is continuous and indicates the desired change of position. The dynamics are stochastic, and the initial state and goal are sampled uniformly at random for each episode. To increase the difficulties of credit assignment and exploration, the agent is always initialized in the lower left corner, and we randomly sampled goal states that are at least 15 steps away. The layout of the obstacles is taken from the classic FourRooms domain, but dilated by a factor of three.

2D Navigation

### E.8 Jaco Reach Environment

We implemented a reaching task using a simulated Jaco robot. Goal states $\psi$ were sampled from uniformly from the interval $[-0.1, 0.1] \times [-0.1, 0.1] \times [0.02, 0.4]$. The agent controlled the velocity of 6 arm joints and 3 finger joints, so the action space was 9 dimensional. The action observation space was 43 dimensional. Let $s_{\mathrm{XYZ}}$ denote the XYZ position of the robot end effector. The reward function was defined as

$$r_\psi(s, a) = \mathbb{1}(\|s_{\mathrm{XYZ}} - \psi\|_2 \leq m) - 1.0$$

and the episode terminated when $\|s_{\mathrm{XYZ}} - \psi\|_2 \leq m$, where $m > 0$ is a margin around the goal. We used $m = 0.1$ and $m = 0.01$ in our experiments. We also reset the environment after 250 steps if the agent had failed to reach the goal.

Jaco Reach

### E.9 Walker Environment

The walker environment was a modified version of the environment from Tassa et al. [50]. We modified the initial state distribution so the agent always started upright, and modified the observation space to include the termination signal as part of the observation. For the linear reward function, the features are the torso height (normalized by subtracting 0.5m), velocity along the forward/aft axis, the XZ displacement of the two feet relative to the agent's center of mass (the agent cannot move along the Y axis), and the squared L2 norm of the actions. The task coefficients $\psi \in \mathbb{R}^d$ can take on values in the range $[-1, 1]$ for all dimensions, except for the control penalty, which takes on values in $[-1, 0]$. Episodes were 100 steps long.

Walker

### E.10 Half-Cheetah Environment

The half-cheetah environment was taken from Tassa et al. [50]. We define tasks to correspond to goal velocities and use the reward function from Rakelly et al. [39]:

$$r_\psi(s, a) = -|s_{\mathrm{vel}} - \psi| - 0.05\|a\|_2^2,$$

where $s_{\mathrm{vel}}$ is the horizontal root velocity. Tasks were sampled uniformly $\psi \in [0, 3]$, with units of meters per second. Episodes were 100 steps long.

Half-Cheetah

### E.11 Desk Environment

The environment provided by Lynch et al. [32] included 19 tasks. We selected the nine most challenging tasks by looking how often a task was accidentally solved. In the demonstrations for each task, we recorded the average return on the remaining 18 tasks. We chose the nine tasks whose average reward was lowest. The nine tasks were three button pushing tasks and six block manipulation tasks.

For experiments in this environment, we found that normalizing the action space was crucial. We computed the coordinate-wise mean and standard deviation of the actions from the demonstrations, and modified the environment

Desk Manipulation

to implicitly normalize actions by subtracting the mean and dividing by the standard deviation. We clipped the action space to $[-1, +1]$, so the agent was only allowed to command actions within one standard deviation (as measured by the expert demos). Another trick that was crucial for RL in this environment was clipping the critic outputs. Since the reward was in $[0, 1]$ and the episode length was capped at 128 steps, we squashed the Q-value predictions with a scaled sigmoid to be in the range $[0, 128]$.

# F   Failed Experiments

1. **100% Relabeling**: When using inverse RL to relabel data for off-policy RL, we initially relabeled 100% of samples from the replay buffer, but found that learning was often worse than doing no relabeling at all. We therefore switched to only 50% relabeling in our experiments. We speculate that retaining some of the originally-commanded goals serves as a sort of hard-negative mining.

2. **Coordinate Ascent on Eq. 6**: We attempted to devise an EM-style algorithm that performed coordinate ascent in Eq. 6, alternating between (1) doing MaxEnt RL and (2) relabeling that data and acquiring the corresponding policy via behavior cloning. While we were unable to get this algorithm to outperform standard MaxEnt RL, we conjecture that this procedure would work with the right choice of inverse RL algorithm.