[Reviews · NeurIPS 2020]

Review 1

Summary and Contributions: EDIT: After reading the rebuttal, my original score stands reinforced. This paper generalizes the idea of hindsight goal relabeling to multi-task settings with a broad class of reward functions by using Maximum Entropy Inverse RL. Further, it derives a link between MaxEnt (forward) RL and MaxEnt Inverse RL. This work provides a principled approach to hindsight relabeling, compared to heuristics common in literature, which also extends its applicability. It also proposes an RL and an Imitation Learning algorithm based on Inverse RL relabeling. Prior relabeling methods can be seen as a special case of the more general algorithms derived here. These algorithms are shown to be significantly more sample efficient than existing methods on a variety of tasks and learn how to solve tasks on which prior methods are inapplicable.

Strengths: + The theory developed in the paper is interesting as it formalizes and generalizes a technique that is typically based on heuristics. Further, the connection between MaxEnt RL and MaxEnt Inverse RL is insightful. + The contributions are novel as far as I am aware. There is a concurrent work [1] cited that arrives upon a similar insight, but the authors clarify the distinction from it. + The methodology in the main paper seems sound, although I did not verify details in the appendix. + The experiments show that the proposed algorithms outperform prior work on goal relabeling by a significant margin. Experiments are conducted on a wide variety of tasks with varying goal conditions. A very thorough evaluation! + Finally, the paper is very clearly written and easy to read! [1] Li, Alexander C., Lerrel Pinto, and Pieter Abbeel. "Generalized Hindsight for Reinforcement Learning." arXiv preprint arXiv:2002.11708 (2020).

Weaknesses: Overall this is a good submission! I do not have any major objections to the method or evaluation. I would be interested to see a comparison of the approach to the concurrent work [1] in the camera-ready version. The error bounds in figure 5 are very difficult to see. I would suggest increasing their brightness.

Correctness: The methods and claims seem correct to the best of my knowledge.

Clarity: The paper is clear and well written! Small typos are present which can be addressed in the camera-ready version. Here are specific suggestions: Line 139: equivalent [to] Line 141: lets us to -> lets us Line 164: distirbution -> distribution

Relation to Prior Work: Related work is adequately discussed and distinction made from prior approachs.

Reproducibility: Yes

Additional Feedback: A couple of more suggestions for improving the writing: 1. The authors identify that random relabeling does surprisingly well (even better than HER for some tasks). Do you have any thoughts on why that is the case? 2. Some of the discussion in the Broader Impact section on the limitations of the approach belongs in the Discussion section. Perhaps in the camera-ready with more page limit the authors can move that up. There is code provided in the supplementary that I have looked at but not evaluated.


Review 2

Summary and Contributions: This work makes a formal connection between inverse reinforcement learning and relabelling experience (ala HER), and introduces two multi-task RL algorithms (one for off-policy RL, one based on BC) derived from this insight, which empirically outperform HER for off-policy RL and traditional behavioral cloning.

Strengths: The core insight of the paper is very strong, intriguing, and makes clear intuitive sense -- that inverse RL can indeed be a fruitful strategy for relabeling experience. There was an ah-ha moment that for me is the sign of a good idea that is simple and semi-obvious in retrospect. The writing is strong (the use of didactic example and the explanation of the importance of the partition function was appreciated), as are the empirical results. The paper left me with fresh ideas and was a pleasure to read.

Weaknesses: The gap from theory to practice was a little jarring -- that is, what is the impact of the assumptions broken in practice so that the algorithm becomes practical to run? And the algorithmic details of how SAC fit into the HIPI algorithm was unclear, which made it hard to understand the exact dynamics of RL+IRL in practice. For example is, what is happening while SAC is exploring -- how does it affect inverse RL? IIUC, if one has never seen a high-performing trajectory for a given task, then even slightly-less-bad performance will appear by the approximation of IRL to be indicative that it was the intended task (if it approximates optimal only through the lens of transitions it has so far seen). That may indeed be a good thing, as it may provide a gradient for the algorithm to potentially improve on that task, but it seems subtle and deserving of explanation -- e.g. an all-knowing IRL algorithm would not make the same inference, and indeed might cause SAC to fail because it would never label the badly-performing task as the intended task (because it is so far from optimal perhaps relative to a simpler task).

Correctness: Yes

Clarity: The paper is generally very well-written. Some algorithmic details were not immediately clear to this reader -- e.g. in the experiments unless I looked at the appendix it was not clear how SAC was being applied to train multi-task networks (and even after looking at the appendix it was not clear how tasks were sampled when doing rollouts -- I assume from a uniform distribution?). Please be more clear about how data collection fits into the HIPI-RL algorithm (i.e. alg 2 does not describe how the replay buffer is filled -- I assume as usual in SAC with rollouts under the current policy -- and I am guessing with tasks uniformly sampled?). IIUC, SAC is not only a multi-task algorithm, and the modifications made to SAC for it to be multi-task seem important to review (using a task-conditioned policy and having some mechanism to sample tasks for rollouts)?

Relation to Prior Work: Yes

Reproducibility: Yes

Additional Feedback: Figures appear out of order in the paper which is I found mildly annoying because typically such figures are in order; I imagine there is some formal convention for whether figures appear in order or not (i.e. whether figure 3 and 4 on the same page should be in order or if it is okay if figure 4 precedes figure 3 on the same page). Whatever that convention is in NeurIPS style, please follow it.


Review 3

Summary and Contributions: This paper addresses the connections between the inverse reinforcement learning and experience relabelling. It also attempts to answer the question: In hindsight, the experience is for what task optimal? This question is answered via inverse reinforcement learning. Subsequently, it proposes to use inverse RL to relabel experience in multi-goal and multi-task settings. In the experiments, the proposed method demonstrates superior performance.

Strengths: The work is based on maximum entropy reinforcement learning and maximum entropy inverse reinforcement leanring. The author derived that the optimal relabelling distribution corresponds to the maximum entropy inverse RL posterior over tasks. The paper derives a lower bound of the improvement in the maximum entropy RL objective. The paper provides theoretical groundings. In the experiments, the proposed method shows better performance compared to HER empirically. The author claims that relabelling with inverse reinforcement learning accelerates learning. The experiments were carried out on various tasks, such as MuJoCo and robotic simulations. The overall performance are good compared to HER with final state relabelling. However, HER with random future state relabelling performs better than HER with final state relabelling, as shown in the original HER paper. It would be more convincing if the author can compare to HER with random future relabelling too. The didactic example helps to understand the method.

Weaknesses: First, for multi-task settings, the method is limited by a set of predefined tasks. Without the set of commanded tasks, the inverse RL part won't work, in my understanding. Secondly, more advanced baselines are missing, such as Hindsight Experience Replay with future random state relabelling and also maximum entropy-based relabelling (see section "Relation to prior work") . It would be more conniving, if the purposed method is compared with strong baselines. Post-rebuttal: After reading the author's rebuttal, I increase my score because my concerns about the future state relabelling and differences with prior works are partially addressed. Although it would be better if the authors can add some experiments to compare with these two priors works.

Correctness: The claims and method are correct. The empirical methodology is correct.

Clarity: The paper is well written.

Relation to Prior Work: The proposed method is based on maximum entropy RL and maximum entropy inverse RL in the multi-goal and multi-task settings. So, how does the method compare to the previous maximum entropy goal-relabelling methods, such as "Maximum Entropy-Regularized Multi-Goal Reinforcement Learning" in ICML 2019 (link of the paper: http://proceedings.mlr.press/v97/zhao19d/zhao19d.pdf) and "Maximum Entropy Gain Exploration for Long Horizon Multi-goal Reinforcement Learning" in ICML 2020 (link of the paper: https://proceedings.icml.cc/static/paper_files/icml/2020/5247-Paper.pdf)? How would the performance comparison look like? And what would be the advantage of the proposed method?

Reproducibility: Yes

Additional Feedback:


Review 4

Summary and Contributions: This paper describes a formal framework for relabeling past experience in the multi-task RL scenario. It uses the framework of MaxEnt RL objective, extend it to multi-task case, and build the connection of the optimal relabelling strategy to the MaxEnt inverse RL. It also proposes tractable algorithm to do the inverse RL and shows its improvement over baselines. The authors provide addition analysis on batch size which is within my expectation. This helps me better understand and verify the correctness of their method. I am prone to make the paper accepted.

Strengths: The motivation is clear and the idea is straightforward. By establishing a formal framework, this work help extends the intuition behind HER to more general multi-task scenario. The algo is evaluated many different task distribution. The authors also conduct some carefully designed experiment to show how each component helps (e.g., adding reward bias, and making reward sparser)

Weaknesses: I don't see much weakness.

Correctness: As far as I am concerned, the claims seem to be all correct.

Clarity: This paper is well-written and easy-to-follow.

Relation to Prior Work: The paper discuss about how it is different from the previous multi-task RL.

Reproducibility: Yes

Additional Feedback: 1. Since the approximated inverse RL relies on the batch size B, I would love to see some analysis on its effect. 2. Some visualization of the transitions and the comparison between original label and relabeled labels would be nice. E.g., I would imagine the transitions that are commonly used in many tasks would be relabeled into somewhat uniform distribution, while the transition distinctive to each task will stay close to the original task labels. 3. Experiments on Atari?

[Author Response · NeurIPS 2020]

We thank all the reviewers for their detailed and thoughtful comments!

**R3**

R3's two main requests are (1) a comparison against a variant of HER that relabels with future states and (2) a discussion
of two prior works.

**Future state relabeling**: We have already compared against the variant of HER that relabels with future states ("Future
State Relabelling" is the green diamonds in Fig 5). This baseline consistently performed worse than random relabeling.
Note that this baseline is not applicable in the setting of more general task distributions (Fig 6).

**Differences with prior work**: Whereas [Zhao 20] and [Pitis 20] focus solely on goal-reaching tasks, our work is
applicable to tasks beyond goal-reaching, such as discrete sets of tasks, linear reward functions, and more general task
distribution (see Fig 6). While both our work and these works all mention "maximum entropy," the actual contributions
are orthogonal and substantially different:

• [Zhao 20] propose a method for prioritizing experience in a replay buffer.
• [Pitis 20] propose a method for sampling goals for exploration.
• We propose a relabeling method for sharing experience across tasks.

The actual algorithms implemented by [Zhao 20] and [Pitis 20] use a combination of previously-introduced relabeling
strategies: [Zhao 20] uses HER and [Pitis 20] uses a combination of "final state", "future state", and "no relabelling".
We have already compared against each of these relabeling strategies in our goal-reaching experiments (Fig 5). Note
that only the "no relabelling" baseline is applicable to tasks beyond goal-reaching (Fig 6). We'll include a discussion of
both papers in the camera-ready version.

**R1**

Thanks for the writing suggestions! We will revise the paper to (1) discuss the concurrent work [Li 20], (2) add a
discussion for why random relabeling works so well, (3) increase the brightness of error bars in Fig 5, (4) move the
discussion limitations from the Broader Impact to the Discussion, and (5) fix the noted typos.

**R2**

**How does HIPI do Exploration?** Our focus in this paper is how to use previously-collected experience to solve
multiple tasks. How that data is collected is largely orthogonal. In our experiments, we simply sample a task from the
prior, $\psi \sim p(\psi)$ and then take actions using corresponding (stochastic) policy, $\pi(a \mid s, \psi)$.

**Writing**: Thanks for the suggestions! We'll (1) clarify how SAC fits into HIPI, (2) clarify how exploration is done, and
(3) fix the figure ordering. We'll also include a discussion of your observation that approximate inverse RL will assume
the best-seen trajectories for some task are optimal for that task.

**R4**

**Effect of batch size**: We ran an additional experiment varying
the batch size used by HIPI-RL on the sparse 2D reacher. Fig. 1
(right) shows that increasing the batch size significantly im-
proves performance, suggesting that better approximate inverse
RL results in better performance. We used a batch size of 32
for the results in the paper, but this experiment suggests that we
could have gotten stronger results by using a larger batch size.

Figure 1: Varying the batch size on sparse 2D reacher

**Visualizing the inferred goals**: The figures below visualize
the inferred goals on the gridworld example (Sec. 6.1). Each
subplot corresponds to a different transition, denoted by the
orange arrow. Dark blue cells denote likely goals, while white cells denote unlikely goals. When the dynamics are
modified so the agent cannot move left (Fig. 3), states to the left of the agent are no longer inferred as likely goals.

Figure 2: Original gridworld.

Figure 3: Modified gridworld where agent cannot move left.

[Meta-Review · NeurIPS 2020]

The reviewers found the paper to be interesting and to make a solid contribution. There are a number of comments in the reviews that can help improve clarity in the camera ready paper.